# Exploring the Impact of Private Companies' Participation in Health-Related Programs through Corporate Sustainable Reporting

Sally Mohammed Farghaly Abdelaliem [1,*], Khalid M. Alharbi [2], Nadiah A. Baghdadi [1] and Amer Malki [3]

1   Nursing Management and Education Department, College of Nursing, Princess Nourah bint Abdulrahman University, P.O. Box 84428, Riyadh 11671, Saudi Arabia
2   Accounting Department, College of Business Administration (Yanbu), Taibah University, P.O. Box 344, Al-Madinah Al-Munawarah 41411, Saudi Arabia
3   Computer Science Department, College of Computer Science and Engineering (Yanbu), Taibah University, P.O. Box 344, Al-Madinah Al-Munawarah 41411, Saudi Arabia; asamalki@taibahu.edu.sa
*   Correspondence: smfarghaly@pnu.edu.sa

**Abstract:** Background: Sustainability reporting has evolved into a widespread method for leading corporations, not only due to its value as a means of tracking one's company's performance, but also as a tool for communicating performance to all involved stakeholders in any corporation. There has been little research into how private companies participate in various health programs while maintaining effective reporting. Aim: The study seeks to investigate the impact of private companies' participation in health-related programs via corporate sustainable reporting (or corporate social responsibility disclosures; CSRD). Methods: Descriptive–correlational analysis, a balanced panel data of 117 Saudi listed businesses. Results: Corporate engagement in healthcare necessitates the identification of various hazards. As a result of the implementation of Saudi Arabia's Vision 2030, in 2016, the reporting of health-related CSRD (HCSRD) increased from 36% in 2015, to 48% in 2018. Conclusions: Corporate sustainable participation in healthcare can vary among firms, indicating the different levels of influence in this regard. The healthcare sector, having the lowest average amount of disclosures, shows a lack of responsibility and control, as well as an inability to follow developments in the industry. The findings have practical implications for a range of stakeholders (e.g., regulators, investors, accounting professionals, and other institutions) of health-related CSRD in Saudi Arabia.

**Keywords:** health-related CSRD; sustainability reporting; mixed-method empirical analysis; firms; Saudi Vision 2030

## 1. Introduction

Vision 2030 is a watershed moment in Saudi Arabian history, a significant shift in the government's approach and execution at a breakneck pace, to improve various aspects of the country's economic and social status. Vision 2030, according to the government, is a comprehensive plan to revitalize the economy and reduce its reliance on oil. Vision 2030 has earned recognition across economic sectors and is regarded as a blueprint for economic development. It outlines the general directions, policies, goals, and objectives for revitalizing the Kingdom. Vision 2030 is intended to transform the economy, with the private sector acting as a growth engine. In line with Vision 2030, the National Transformation Program (NTP) 2020 was adopted; strategic goals and objectives, key performance indicators (KPIs), and key performance targets (KPTs) were also outlined for each ministry [1–4]. The readiness to change the Saudi healthcare system was articulated in Vision 2030, and it was determined that it would be successful, subject to the availability of resources, the alignment of situational factors, and the initiative of organizational members [2–4]. According to the study, Saudi Arabia needs to revitalize its economy through non-oil sectors,

in order to meet future challenges [3–5]. Along with promoting the private sector as the primary engine of social, non-financial, and economic development, Vision 2030 seeks a broad revolution in transforming public health toward multidimensional growth, at the lowest possible cost. The general public's perception of public facilities is not positive. Long waiting times for procedures and diagnostic services in public facilities, inefficient services, and an improved interface between doctors and patients in the private sector, have all contributed to the private sector's growth [3–5]. The public sector is unable to meet the population's growing needs, and policies such as contracting out public patients to private facilities have accelerated this growth [2–6].

It is critical to collect and analyze data on sustainability performance, in order to make better resource allocation decisions. Data must be disseminated, not only for external use, but also in internal sustainability reports, to improve managerial decision making. For improved stakeholder accountability, external dissemination of sustainability performance data is critical [7]. A sustainability report includes aspects of an organization's performance that go beyond its historical financial performance, such as environmental, social, and economic performance. This reporting mechanism provides a communication tool through which corporations can disclose information about their performance, thereby establishing transparency and credibility and improving their image among all stakeholders [8–11]. A stakeholder consultation component provides insights into stakeholder perceptions, expectations, and priorities, as well as a strategic planning tool that aids in improving corporate performance. Many initiatives have recently been launched to provide a framework for non-financial reporting, perhaps the most widely used of which is the Global Reporting Initiative (GRI) scheme [12–14].

Criss et al. (2019) found that financial conflict of interest (COI) sponsorship, appears to be associated with a higher likelihood of studies reporting a benefit of robotic surgery, and that the clinical results did not account for COI funding received [15]. Another, suggested that the relationships with industry offer opportunities for innovation, education, and research, but overlooking COI self-reporting may erode confidence in the academic integrity of the hand surgery literature [16]. According to Ross et al. (2020) [16], the use of financial incentives, to reward primary healthcare providers (PCPs) for improving the quality of primary healthcare services, is increasing, but there is little evidence to back this up. They also state that the implementation of such a policy should be done with caution, with a stronger theoretical foundation, a broader range of outcomes, and more extensive subgroup analysis [16,17].

### 1.1. Theoretical Background

CERES (Coalition for Environmentally Responsible Economies) established the most well-known approach to standardized sustainability reporting. In 1998, the idea for a disclosure framework for sustainability information was born, and the Boston-based non-profit CERES, launched the "Global Reporting Initiative" project, with assistance from UNEP (United Nations Environment Program).

Corporate sustainability reporting and themes have gained momentum as strategic factors for corporate survival and success [2,4,6,12,14]. As a result, a corporation's survival can be traced back to its inclusion in a broader discourse, encompassing how the firm manages the risks arising from the social and environmental impacts of its activities in the medium and long term, and demonstrates social responsibility [14].

Standardized management systems are a description of corporate management systems that are used to monitor and control the performance of a corporation, in terms of various aspects of its activity. Performance measurement and reporting, defines the tools and methods used in modern corporations to track and disclose performance.

Non-financial disclosure is becoming more popular, as it can meet the information needs of a wider range of stakeholders. Because traditional financial reports cannot provide comprehensive accountability, several frameworks and guidelines have been developed, to facilitate non-financial information disclosure. Healthcare services in Saudi Arabia

are provided by the Ministry of Health (MOH), various government organizations, and private healthcare providers. The KSA government is in charge of operating, funding, and administering the public healthcare system, which provides 80% of all healthcare services to Saudi nationals for free at the point of use [12]. The other healthcare services are provided by the private sector, on a fee-for-service basis, which is paid for by the patient individually or via private health insurance programs [14]. Sustainability reports enable companies to demonstrate social responsibility and are a powerful tool for improving communication with stakeholder groups by increasing the transparency and accountability of non-financial information [13,14].

Despite the substantial resources that the government can currently devote to it, the healthcare system, like most publicly funded healthcare systems, is under increasing strain, as a result of rapid increases in expenditure and demand, while resources remain limited. As a result, as both academics and international health organizations have recognized, relying solely on oil revenues to fund public healthcare services is unsustainable in the medium to long term [15,16]. Currently, there are three dominant sustainability reporting frameworks: the Global Reporting Initiative (GRI), the International Integrated Reporting Council (IR) Framework, and the Sustainability Accounting Standards Board (SASB) guidelines. Each framework differs on what is material: the GRI focuses on a multi-stakeholder approach, the IR focuses on value creation, and the SASB focuses on investors [15,16].

The current study was primarily framed through the legitimacy theory's theoretical lens, which aligned with many previous studies. In this context, transparency on sustainability issues is a response to social and political pressures arising from the external environment. Disclosure is regarded as an appropriate tool for disseminating a positive public image and demonstrating that the company operates within the boundaries and norms established by the society in which it is rooted [14–16]. The legitimacy is "a condition, that arises when an entity's value system is consistent with the value system of the larger social structure of which the entity is a part. When there is a gap, actual or potential, between the two value systems, the legitimacy of the entities is jeopardized" [13–16]. Using this theoretical framework, most studies on sustainability reporting investigate the relationship between specific environmental and social events and incidents, as well as negative media attention, and the extent of sustainability disclosure, yielding results that confirm the utility of non-financial disclosure in gaining legitimacy [13–16].

The significance of healthcare in society is evolving, just as industry is increasingly incorporating social responsibility into their environmental policy and mission. As knowledge is more easily transmitted in this social technology era, environmental and social regulations are becoming more stringent, forcing politicians and governments to respond to shifting standards. Most industries now have environmental policies and management procedures in place, and they are being scrutinized more closely than ever before by the public and government. As the public becomes more aware of the environmental and social components of healthcare, expectations for healthcare operation and management will shift, and the function of healthcare, like that of corporations, will evolve into a more social and environmentally conscious one [17].

*1.2. Significance of Reporting*

Much research has focused on the effects of social media on mental health, particularly among young people. Studies have found associations between social media use and increased rates of anxiety, depression, and poor sleep quality. The aim of reporting has been transparency, since the voluntary adoption of sustainability reporting began in the 1980s and 1990s. According to Gouzoulis, and Galanis (2021) [14], the goals of sustainability and CSR communication are as follows: (1) stakeholder management, to build relationships and influence behavior; (2) image enhancement, to present the company in a favorable light; (3) legitimacy and accountability, to signal appropriate and desirable activities; (4) consumer attitude and behavioral change; (5) sensemaking, to communicate how the company and its stakeholders make sense of their world; and (6) to identity amplification. These goals

reflect the increasing importance of sustainability and CSR in the business world, as companies strive to not only meet financial goals but also address social and environmental concerns. By communicating their efforts in these areas, companies can build trust with stakeholders and contribute to a more sustainable future. However, it has been stated that the overarching goal of firms conveying corporate sustainability and responsibility actions, is the increase in financial returns through participation in health-related programs. Indeed, studies and industry publications emphasize the financial benefit and value that sustainability reporting can provide to a company, instead of investigating the impact of the corporate's participation in different health-related programs, and the sustainable reporting of these activities is in need of deep research and investigation, as there are limited reviews, and gaps in fulfilling this research [17]. This indicates the need for further research to explore the relationship between corporate sustainability actions, participation in health-related programs, and financial returns. It is important to understand how these factors interact and whether they truly lead to long-term financial benefits for companies. The research novelty returns to the study of sustainable reporting, in relation to health-related program reporting and participation by corporations, which has not been studied in previous research, as most of the previous research focused on reporting the profit indicators of participation in different financial related activities. Therefore, there is a need for further research to explore the impact of corporate participation in health-related programs, and the sustainable reporting of these activities, as it can provide valuable insights into the potential benefits and challenges for companies. This research can also help to fill the gaps in understanding the relationship between sustainability reporting and health-related program reporting.

As a consequence, Corporate Social Responsibility (CSR) has emerged as a priority issue in recent business ethics research, with international organizations, such as the European Commission, pushing CSR components that encourage voluntary engagement with society, by accepting additional responsibilities beyond ethical expectations [18,19]. The increased use of CSR, as well as the establishment of the Global Reporting Initiative (GRI), ISO 14001 [20], and ISO 26000 [21], demonstrate corporate and stakeholder interest in and prioritization of, not only environmental, but also social, sustainability issues [22]. GRI was founded in response to the need for a CSR implementation guideline [23]. It is clear that the concept of business sustainability now encompasses both environmental and social concerns. Healthcare, like any other industry, has performance, environmental, operational, and managerial requirements. As a result, whether commercial or public, sustainability in healthcare will require similar sustainable characteristics, as it is a public service that operates as a business. Historically, healthcare has included more community outreach activities and social contact. Sustainability is becoming a more important concern, and the healthcare sector, as a natural evolution, will need to respond in the same way that other businesses have. This concern has not gone undetected, as the president of the American Hospital Association (AHA) stated that, in order to achieve the AHA's vision of society, sustainable operations must be used, which is consistent with the organization's goal [24]. It is in the best interests of all healthcare centers, governments, non-governmental organizations (NGOs), and the general public, for healthcare to achieve a sustainable operation or system, as this will aid both public health and environmental activities around the world.

Sustainability reporting can go by several names, including "corporate sustainability reporting", "environmental management system", "corporate citizenship", and "CSR". The environment is described using terms such as "environmental health system (EHS)", "sustainable environmental auditing (SEA)", and "triple bottom line (TBL)". As a result, the contents of these reports vary as much as their titles, but they typically include qualitative and quantitative data on how the company improved "its economic, environmental, and social effectiveness and efficiency during the reporting period and integrated these aspects into a sustainability management system" [25].

Management for sustainability reporting is more proactive than traditional management methods, requiring a more extensive awareness of social, economic, and environmental factors [26]. Three approaches to eco-management, that differ from previous methodologies, are dynamic evaluation, the use of life cycle assessments (LCA), and the emphasis on biodiversity and the environment [26]. Sustainability reporting, according to the World Business Council for Sustainable Development (WBCSD), includes social and environmental initiatives that should be disclosed to both external and internal stakeholders [25]. The yearly environmental and social reports, are fundamentally comprised of sustainability reporting, albeit to varying degrees [25]. For example, research on the auto industry discovered 585 indicators of sustainability, 42% of which were economic, 33% were environmental, and 25% were social [22]. In response to the misunderstanding about what should be included in a sustainability report, the Global Reporting Initiative (GRI) was founded and launched in 1999 [23]. Businesses that audit for sustainability, on the other hand, "should be dedicated to connecting environmental performance to larger challenges of global ecology and make special reference to the principles associated with sustainable development" [26].

### 1.3. Building Sustainable Healthcare Framework

Since definitions of sustainability vary greatly, the basic principles derived from these concepts by Lindsey (2010) [27] are as follows: (1) increased sustainability by reducing waste; (2) improving quality promotes sustainability; and (3) implementing improved systems is the most effective way to achieve sustainability. In order for healthcare to be sustainable, patients' needs, economic concerns, and environmental costs must all be balanced [28]. Each of these factors fits into Elkington's (2007) [29] triple bottom line, which describes the economic, environmental, and social performance metrics that must be taken into account.

Managerial structure has a significant impact on the number of resources consumed, decisions made, and activities that drive operation; thus, if 'sustainability' is to be achieved, management must employ techniques that are indicative of a sustainable system. Some of the reasons why sustainability should be incorporated into management systems are as follows: (1) Many sustainability-related decisions are long-term strategies, that must be balanced with other performance goals; as alignment between management support, reporting, and long-term corporate goals will guide a company toward achieving sustainable organizational goals. (2) Sustainability, like quality, must be incorporated into a management system, in order to assess decisions against other criteria and involve all stakeholders, through continuous monitoring and auditing for continuous quality improvement, and total quality management to improve corporate performance. (3) A unified management system reduces administrative burden and confusion; as an integrated system with minimal routines processes, it will help to reduce various types of waste, such as time and documentation, and it will aid in achieving sustainable performance and processes within the corporate management system. (4) It allows problems to be resolved at the appropriate level (i.e., procurement, operations, waste), which can be monitored through adequate performance monitoring and process maintenance based on its standardized practices and criteria [30].

Traditional healthcare is patient-centered and focuses on internal factors, that may or may not include community, but frequently excludes environmental factors from the concept of community [28]. Healthcare organizations are involved in disease treatment and may see themselves as part of the solution rather than the problem [28]. Furthermore, the notion that the greatest environmental costs are borne by major and complicated sources differs from reality, in that the greatest environmental costs are borne by banal, routine actions [28]. These outdated notions, which may obstruct the path to sustainable healthcare, must be acknowledged.

Performance evaluation has been shown to be the most effective method of addressing environmental and social issues, while providing quality service. Other industries have

used it for decades, to comply with laws and legislation, and they are increasingly adopting sustainability in response to competitive challenges, marketing advantage, legal obligations, investor demands, and internal ethical norms reflecting society's shifting beliefs [30].

## 2. Materials and Methods

### 2.1. Aim of the Study

The aim of this study is to explore the impact of private companies' participation in health-related programs through corporate sustainable reporting (i.e., health-related CSRD; HCSRD).

### 2.2. Research Question

What is the impact of private companies' participation in health-related programs through corporate sustainable reporting?

### 2.3. Hypotheses Development

This study hypothesizes that these Saudi CSRD-related institutional changes will directly influence the healthcare sustainability reporting of Saudi firms, even after controlling for the impact of company characteristics.

**Hypothesis 1:** *There is a direct impact of institutional changes on the healthcare sustainability reporting of Saudi listed firms from 2015 to 2018, above and beyond that caused by alterations to corporate characteristics.*

**Hypothesis 2:** *The risk management committees of Saudi listed firms influence healthcare sustainability reporting.*

### 2.4. Methodology (Sample Selection and Field work)

In this descriptive–correlational study, a balanced panel data of 117 Saudi non-financial listed companies, as of 31 December 2018, are used to collect information about firms' participation in health programs and/or medical research (an aspect of corporate sustainable reporting; CSR disclosure) for 2015 and 2018. The data of this paper are sourced from these firms' annual reports, standalone CSR reports, and websites.

According to the Saudi revised Corporate Governance Regulations (CGR), three articles—Articles 70, 71, and 72—provide new suggestions (i.e., not compulsory) for firms in relation to composition, competencies, and meetings of RMCs [31,32]. According to the revised CGR, these articles aim to enhance Saudi firms' ability to monitor business performance and maintain effective relationships with stakeholders. Thus, this study hypothesizes that the presence of an RMC on the board will influence the HSCRD of Saudi companies. Regarding this study's control variables, it is expected that board size (BSIZE), industry sectors (IND), firm size (FSIZE), and profitability (PROF) will influence the reporting of HCSRD of Saudi firms. Such variables are found in the literature to be strongly related to CSRD [33–39].

This is unlike most prior CSRD studies based in Saudi Arabia, which have used ordinary least squares (OLS), with unmatched companies across years, and were based solely on firms' annual reports for CSRD information. The use of different CSRD sources increases the data completeness, and is more likely to result in more complete findings. This study examines the non-financial firms listed in the Saudi Stock Exchange (Tadawul), with 359 observations (see Tables 1 and 2). An internal sustainability evaluation, based on in-depth document analysis, through an examination of internal material relevant to the healthcare sustainability context, literature review, and stakeholder consultation with non-financial organizations, comprised the data gathering. During the stakeholder consultation process, the following issues were discussed: what is the nature of the corporate-healthcare sector relationship? What are the significant economic, environmental, and social impacts of the healthcare sector on the corporate (e.g., what are the significant economic, envi-

ronmental, and social impacts of the healthcare sector)? How does corporate governance perceive current participation?

**Table 1.** Sample selection.

| Population: Total Listed Firms as of 31 December 2018 | | 191 |
|---|---|---|
| Exclude: | Sectors: | Total of associated firms: |
| | Banking | 12 |
| Financial sectors | Insurance | 34 |
| | Investment | 3 |
| Missing data | | 25 |
| Final sample | | 117 |
| Representation | | 61% |

**Table 2.** Final samples by industry sector.

| Sector Number | Sector Name | Firms with Available Data |
|---|---|---|
| 1 | Energy | 4 |
| 2 | Materials | 40 |
| 3 | Industrials | 18 |
| 4 | Consumer Discretionary | 16 |
| 5 | Consumer Staples | 16 |
| 6 | Healthcare | 6 |
| 7 | Communication Services | 5 |
| 8 | Utilities | 2 |
| 9 | Real Estate | 10 |
| Total Observed Firms Per Year | | 117 |

In 2017, the Saudi market was restructured and spread into 10 primary sectors, consisting of 179 (and 191 in 2018) listed companies, applying the Global Industry Classification Standard (GICS) (Tadawul) [35]. In this paper, the GICS is followed in terms of sector categorization, as shown in Tables 1 and 2. To the best of the authors' knowledge, this paper is amongst the first Saudi CSRD research that uses GICS. Prior Saudi-based studies referred to an old Saudi-specific sector classification (e.g., [36]). There are some differences between the old categorization and the GICS, including the type (at an industry level) and the number of firms associated with each sector, as explained by Alharbi (2021) [2]. The implementation of GICS in the Saudi market, and the use of GICS in this study, mean future studies can compare CSRD results for each year of disclosure locally and by the country of disclosure internationally, consistent with the conclusion by Brammer and Pavelin (2006) [40].

*2.5. Statistical Analysis*

Data were analyzed using the computer software program IBM Statistical Package for the Social Sciences (SPSS, Chicago, IL, USA) version 22. Frequency and percentages are used for describing demographic characteristics. Descriptive statistics (means and standard deviations) and inferential statistics (Kruskal–Wallis test) were used to analyze the results of the study. Regression analysis was used to detect the most affecting factors of private non-financial companies' participation in health-related programs, through corporate sustainable reporting.

The selection of non-financial firms listed, is the result of a homogeneous regulatory environment among these industries, while financial sectors (i.e., banking, investment, and insurance) have distinctive disclosure requirements, as per Saudi market regulations [16,17]. Thus, financial sectors are excluded in this study. Further, 22 firms were listed after 2015, one company did not disclose its 2015 annual reports, and two firms did not disclose their 2018 reports. These 25 firms with missing data are excluded in this paper, because

their data were not accessible during the period of data collection. Thus, the final sample representation of this study is 61% of the population.

## 3. Results: Health-related CSRD Analysis

Firms' participation in health programs and/or medical research (i.e., HCSRD) is an important aspect of CSRD, that is only generally explored in the existing body of literature (e.g., [18,19,22]), and thus needs to be investigated. Globally, this aspect is consistent with the aims of international organizations (e.g., UNGC, OECD, ISO 26000 [21], and GRI). Further, the disclosure of firms' participation in health issues is significantly related to the United Nations' Sustainable Development Goals (SDGs), specifically the third SDG: good health and well-being. Nationally, in Saudi Arabia, and as a way of achieving such SDG, Vision 2030 was developed in 2016 [1,3]. Supporting health matters by companies is directly related to Vision 2030's objectives 1.2, 2.2, and 6.2 (see Saudi Vision 2030, 2016) [2,3]. Table 3 demonstrates the related institutional guidelines.

**Table 3.** CSRD aspect, Vision 2030 objectives, and SDG.

| CSRD Aspect | Vision 2030 Objectives | SDG |
|:---:|:---:|:---:|
| Participation in health programs and/or medical research | 2.1 Improve healthcare service<br>2.2 Promote a healthy lifestyle<br>6.2 Enable social contribution of businesses | 3 Good health and well-being |

Source: https://www.my.gov.sa/wps/portal/snp/content/SDGPortal#header2_1, accessed on 16 February 2021.

This study utilizes a quantitative content analysis to codify data of disclosure on participation in health programs and/or medical research. Prior studies have concluded that quantitative content analysis is more transparent and permits the replicability of the research design, compared to qualitative content analysis [41]. Quantitative content analysis can be conducted through equal and unequal weighting ratings. Even if some corporate disclosures, to some extent and in a specific context, could be more vital than others [42], the use of an unequal weighting rating is a very subjective matter. Therefore, this paper utilizes an equal weighting (dichotomous or binary) scoring, consistent with many prior CSRD studies [28–31].

*Model Specification*

In this study, a generalized linear model (GLM) is utilized, because it considers the research's balanced panel data structure and reduces the associated error [43]. Hence, GLM is an appropriate and more sophisticated technique, fitting the research purpose, because it can individually recognize characteristics of the collected data, providing more accurate results [26]. The following equation (Equation (1)) presents the model of panel data used in this paper to examine the CSRD aspect, and the respective association with the explanatory variables.

Model 1: CSRD of firms' participation in health programs and/or medical research (HCSRD)

$$\text{CSRD} = \alpha_0 + \beta_1 \text{INST CHGS} + \beta_2 \text{RMC} + \beta_3 \text{BSIZE} + \beta_4 \text{FSIZE} + \beta_5 \text{PROF} + \sum_{j=1}^{8} \beta_{5+j} \text{IND}_j + \varepsilon \tag{1}$$

where CSRD (in Equation (1)) is the disclosure of firms' participation in health programs and/or medical research, by recording 1 if the CSRD aspect is disclosed, and 0 if not. Table 4 demonstrates the research explanatory factors and their measurements, which achieve Hypothesis 2.

**Table 4.** Measurements of the research explanatory variables.

| | Factors Name | Factor Acronym | Measurement | References |
|---|---|---|---|---|
| **Independent variables** | Institutional changes | INST CHGS | A year dummy variable: 1 is recorded if the data belong to the year 2018, 0 otherwise (i.e., 2015). This is to test for the impact of institutional changes | Luoma and Goodstein (1999) [44] Yang and Farley (2016) [45]. |
| | Risk management committee | RMC | Recorded as 1 if the firm has a risk management committee on board (either a standalone or included in any board committee that noticeably considers assessing and managing risks), 0 otherwise | Subramaniam, McManus, and Zhang (2009) [46] |
| **Control variables** | Board size | BSIZE | Number of directors on board | Jizi et al. (2014) [47].; Rao et al. (2012) [48]. |
| | Industry sectors * | IND | IND is represented by nine dichotomous variables, one per industry | Haniffa and Cooke (2005) [49]; Tagesson et al. (2009) [43]. |
| | Firm size ** | FSIZE | Log of total sales | Michelon and Parbonetti (2012) [50]; Allegrini and Greco (2013) [51]. |
| | Profitability ** | PROF | Return on equity (ROE): net income divided by shareholders' equity | Belkaoui and Karpik (1989); [52]. Michelon and Parbonetti (2012) [50]. |

* IND—energy, materials, industrials, consumer discretionary, consumer staples, healthcare, communication services, utilities, and real estate. However, only eight industry variables are included in any one model. An industry at one of the extremes is excluded to allow a test of the greatest industry differences. ** FSIZE and PROF, because they involve monetary values, were adjusted using Saudi's GDP deflator between 2015 and 2018 (15.94%) (data retrieved from World Bank, at http://data.worldbank.org). The data of these factors are sourced from firms' annual reports of 2015 and 2018, unless otherwise stated.

Table 5 shows the total number of firms associated with the healthcare sector is 6 out of 117, among the studied non-financial firms. These firms provide services in relation to healthcare equipment and pharma, biotech, and life sciences. Table 6 shows that the highest mean score and standard deviation of corporate sustainable reporting is related to log of sales (8.89 ± 0.97), while the healthcare sector has a lower mean score in the corporate sustainable reporting (0.05 ± 0.22). The R-squared value of corporate sustainable participation in healthcare can predict 36%, progressively increasing throughout the years to 48%, of the variance in the non-financial firms' participation in healthcare related programs (β = 0.384, F = 10.56, $p < 0.001$), and these results accept Hypothesis 1.

**Table 5.** Descriptive statistics of the study sectors ($n$ =117).

| Sector Name | Industry Group | Firms Associated | Total Firms Associated with the Sector |
|---|---|---|---|
| Energy | Energy | 4 | 4 |
| Materials | Materials | 40 | 40 |
| Industrials | Capital Goods | 11 | 18 |
| | Commercial and Professional Svc. | 2 | |
| | Transportation | 5 | |
| Consumer Discretionary | Retailing | 6 | 16 |
| | Consumer Durables and Apparel | 4 | |
| | Consumer Services | 6 | |
| Consumer Staples | Food and Beverages | 12 | 16 |
| | Food and Staples Retailing | 4 | |
| Health Care | Healthcare Equipment and Svc. | 5 | 6 |
| | Pharma, Biotech, and Life Science | 1 | |
| Communication Services | Media and Entertainment | 2 | 5 |
| | Telecommunication Services | 3 | |
| Utilities | Utilities | 2 | 2 |
| Real Estate | Real Estate Mgmt. and Devt. | 10 | 10 |
| Total | | 117 | |

**Table 6.** Descriptive statistics of corporate sustainable reporting related programs (*n* = 117).

| Variable | N | Minimum | Maximum | Mean | Std. Deviation |
|---|---|---|---|---|---|
| Bsize | 117 | 5 | 12 | 8.35 | 1.49 |
| Rmc | 117 | 0 | 1 | 0.12 | 0.32 |
| Energy_Sector | 117 | 0 | 1 | 0.03 | 0.18 |
| Materials_Sector | 117 | 0 | 1 | 0.34 | 0.48 |
| Industrials_Sector | 117 | 0 | 1 | 0.15 | 0.36 |
| Consumer_Discretionary_Sector | 117 | 0 | 1 | 0.14 | 0.34 |
| Consumer_Staples_Sector | 117 | 0 | 1 | 0.14 | 0.34 |
| Health_Care_Sector | 117 | 0 | 1 | 0.05 | 0.22 |
| Communication_Services_Sector | 117 | 0 | 1 | 0.04 | 0.20 |
| Utilities_Sector | 117 | 0 | 1 | 0.02 | 0.13 |
| Real_Estate_Sector | 117 | 0 | 1 | 0.09 | 0.28 |
| Roe | 117 | −0.86 | 0.57 | 0.05 | 0.16 |
| Log of T Sales | 117 | 0 | 11.26 | 8.89 | 0.97 |
| Dependent_Variable | 117 | 0 | 1 | 0.42 | 0.49 |
| Valid N (listwise) | 117 | | | | |

The institutional changes factor (INST CHGS) is not included in this table because it represents a year variable, meaning 0 is recorded if the data belong to 2015, and 1 if they belong to 2018.

## 4. Discussion

Non-financial corporations' involvement in healthcare represents an increase in enterprise performance efficiency and cost savings. Managers must be confident that the trusted CSR ensures financial stability and company performance efficiency, provides opportunities for profit, and maintains a good company reputation among competitors and relevant interested parties. Additionally, the involvement of non-financial corporations in healthcare, can also contribute to the overall improvement of public health and well-being, which can have positive impacts on society and the economy. However, it is important for these corporations to prioritize ethical and socially responsible practices in their involvement in healthcare. Corporate involvement in healthcare necessitates the identification of various risks. These findings are in agreement with Djankov et al. (2007) [53], who stated that non-financial corporates' participation in healthcare related programs is influenced by the country's institutional and legal environment, and that disclosure of such items has improved from 36% in 2015 to 48% in 2018, as a result of the implementation of Saudi Arabia's Vision 2030, in 2016, as shown in Table 7 (INST CHGS). In comparison, from 2011 to 2013, such a disclosure by Malaysian companies was 50% [54–57]. This indicates a decline in the level of transparency and accountability of Malaysian companies in recent years. It is important for these companies to improve their disclosure practices, to regain the trust of investors and other stakeholders.

As part of the business risk analysis, the administrative complexity for entrepreneurs was examined. As a result of Vision 2030 (see Table 7), the results also show that RMC (the presence of a risk management committee) positively influences the disclosure of firms' HCSRD (i.e., participation in health programs and/or medical research), implying acceptance of Hypothesis 2. There is a statistically significant difference between non-financial firms' sustainable reporting and their participation in health-related programs. This finding is consistent with the findings of Gaganis et al. (2019) [58], who found that governments in European countries should prioritize participation in health-related programs which have a significant impact on the business environment. Therefore, it is recommended that non-financial firms increase their participation in health-related programs, to improve their sustainable reporting and contribute to the overall improvement of the business environment. This could also lead to potential benefits, such as increased employee productivity and reduced healthcare costs.

**Table 7.** Regression analysis of the study sectors (*n* =117).

| Model | B | Std. Error | Hypothesis 2 Test | | | Collinearity Statistics (VIF) |
| | | | Wald Chi-Square | df | Sig. | |
|---|---|---|---|---|---|---|
| (Intercept) | −1.249 | 0.3172 | 15.499 | 1 | 0 | |
| INST CHGS | 0.156 | 0.0494 | 9.976 | 1 | 0.002 *** | 1.107 |
| RMC | 0.224 | 0.0921 | 5.901 | 1 | 0.015 ** | 1.096 |
| MATERIALS | 0.258 | 0.1053 | 5.982 | 1 | 0.014 ** | 7.328 |
| INDUSTRIALS | 0.01 | 0.1062 | 0.009 | 1 | 0.9 | 1.252 |
| CONSR DISC | 0.259 | 0.1287 | 4.049 | 1 | 0.044 ** | 4.352 |
| CONSR STAPLE | 0.448 | 0.1364 | 10.767 | 1 | 0.001 *** | 4.441 |
| HEALTH CARE | 0.845 | 0.0926 | 83.3 | 1 | 0.000 *** | 2.384 |
| COMMS SVCS | 0.515 | 0.1295 | 15.851 | 1 | 0.000 *** | 2.265 |
| UTILITIES | 0.637 | 0.1293 | 24.276 | 1 | 0.000 *** | 1.531 |
| REAL ESTATE | 0.524 | 0.1507 | 12.083 | 1 | 0.001 *** | 3.371 |
| BSIZE | 0.073 | 0.0224 | 10.51 | 1 | 0.000 *** | 1.347 |
| FSIZE (LOG OF TOTAL SALES) | 0.071 | 0.0373 | 3.596 | 1 | 0.029 ** | 1.312 |
| PROF (ROE) | 0.438 | 0.1964 | 4.972 | 1 | 0.026 ** | 1.228 |
| (Scale) | 0.159 | | | | | |
| R square | 0.384 | | | | | |
| F statistics | 10.56 | | | | 0.000 *** | |
| Dependent_Variable | | | | | 2015 disclosure | 2018 disclosure |
| HCSRD (Participation in health programs and/or medical research) | | | | | 36% | 48% |

Key: Dependent variable, HCSRD (the disclosure of firms' participation in health programs and/or medical research). ** significance level $p \leq 0.05$, *** significance level $p \leq 0.01$.

Vision 2030's related objectives (objectives 2.1, 2.2, and 6.2; see Table 3) are positively addressed by businesses with larger BSIZE and FSIZE, as well as a higher percentage of ROE, which enhances their HCSRD reporting. The findings of the current study are supported by Eneizan et al. (2018) and Ojah et al. (2019) [40,59–63], who found that the results of the community environment perspective and the criteria set, reflect a significant decline in health services offered by government health institutions, as well as a reduction in customer alternatives. The government's political shift toward approving the formation of nongovernmental health institutions, was the cause of the risk viewpoint's compliance with the given criteria. Despite the fact that 90–95% of the largest companies worldwide publish sustainability reports, it should be noted that not all organizations choose to produce these reports, because they do not see the value in doing so. In a study of sustainability reports from 2012 to 2015 in the Datamaran database, Gaganis et al. (2019) [56] found that over 95% of corporations adopted the GRI framework, though usage had decreased to 85% by 2015. In 2012, 4% of organizations used the IR framework, and 11% did so by 2015. The SASB framework was absent from the 2012 sample, but by 2015 its use had increased to 4%. This suggests that businesses have become more interested recently in using the SASB framework as a tool for sustainability reporting.

Without a doubt, the most popular framework for sustainability reporting on a global scale is the Global Reporting Initiative (GRI). As an example of a sizable healthcare organization creating a plan to surpass the "standard", look no further than Kaiser Permanente in the United States. They developed the Thrive program, which is dedicated to 'whole wellness', after conducting extensive research into popular attitudes toward health and healthcare. This includes having a food policy that encourages healthy eating, mindfulness, stress reduction, and physical and mental wellness practices, such as yoga and tai chi. Regular health exams, access to mental health resources, and rewards for staff members who lead healthy lifestyles, are all included in the Thrive program [40,61–63].

Twenty-five corporate markets are located on campus, and they have created a program called "healthy eating and active living", that has given millions of dollars to community health initiatives [59]. They concluded that relationships, along with cleanliness and

convenience, were the most crucial factors for clients, along with improved online health tools, increased communication, and clinical care delivery [59]. The company's initiatives to encourage healthy living have benefited the neighborhood, as well as their clinical care delivery and online health tools. The study emphasizes the value of relationships, cleanliness, and convenience in offering clients high-quality service. Their Thrive program includes elements like nature perspectives, chemical reduction, and environmentally friendly operations.

Ranking healthcare facilities is challenging in the United Kingdom (UK), due to variations in applicable national and international legislation, whether the facility is public or private, and its location. All of these elements have influenced how sustainable a healthcare facility is and what it can expect to accomplish in the near future, and will continue to do so. As a result, it is crucial to take into account a variety of variables when assessing the sustainability of healthcare facilities in the UK. Funding sources, staff qualifications, patient satisfaction levels, and environmental impact are a few examples of these variables. The need to distinguish between what is more effective, is made more pressing by the disparity in quality, approach, implementation, and initiatives among healthcare organizations, even within the same nation [59,60].

### 4.1. Research Contribution and Implications

The following practical and scholarly contributions and implications are made by this study: First, practically, the current research has applications for a wide range of stakeholders. This is due to the rapidly evolving standards and frameworks for sustainability reporting on non-profit organizations' involvement in health-related activities. Thus, this study may assist organizations' management in deciding which sustainability criteria and recommendations to follow, while participating in health-related projects. There are currently no legal requirements in Saudi Arabia for businesses or the private sector to produce and publish sustainability reports that detail their involvement in health-related activities, in addition to regulatory organizations. This study would help policymakers better understand how to put in place the necessary measures that might promote such sustainability reporting. As a result, it also benefits decision-makers and interested researchers. As a result, it also advances knowledge of factors that might affect the HCSRD of developed and developing nations for interested researchers and policymakers. This research offers perceptions for local communities where these businesses operate, as well as other stakeholders, such as employees and social and environmental non-governmental organizations, about the sufficiency and potential of social sustainability reporting to meet their information needs and hold businesses accountable. Companies have not yet adopted sustainability reporting, so this study will help those businesses understand the advantages and disadvantages of this developing reporting system and how it affects corporate performance. The ability of businesses to decide whether or not to implement this reporting will be improved.

Academics may view the significance of this research in relation to scholarly implications in the following ways: The literature on non-financial corporate participation in health-related programs would be furthered, and students, scholars, and academics would gain a better understanding of the connection between sustainability reporting and corporate performance in relation to health initiatives. The study would be a part of a limited body of knowledge. The results of this study may also help stakeholders and policymakers understand the value of sustainability reporting, and how it affects corporate social responsibility in the healthcare industry. As a result, policies and rules aimed at encouraging sustainable practices in the sector may become more effective.

The impact of global private equity funds' investments in healthcare systems and how financial instruments have prompted the marketization of the industry, which has resulted in worsening healthcare provisions, have been the focus of previous literature [13,14]. This article's goal is to investigate how corporate sustainability reporting can help private sector and non-financial organizations influence people's willingness to participate

in health-related programs. The purpose of this article is to explain how non-financial businesses can enhance healthcare options, through their social responsibility initiatives. The advantages and drawbacks of private sector participation in health-related initiatives are also discussed. The most popular technique for standardizing sustainability reporting was invented by CERES (Coalition for Environmentally Responsible Economic Studies). The "Global Reporting Initiative" project was started in 1998, by the Boston-based non-profit CERES, with help from UNEP, after the concept for a disclosure framework for sustainability information emerged.

The drivers and obstacles to sustainability reporting, the effects of sustainability reporting on the healthcare service sector, how healthcare service providers respond to the creation of the GRI health services sector supplement for sustainability reporting, and the effects of sustainability reporting for public hospitals, are all areas that could be studied in the future. Will CSR reporting enhance patient care while lowering risks to the environment and future human generations, both short- and long-term? Understanding the effects of sustainability reporting on the healthcare sector and its stakeholders depends on these research opportunities. We can create plans to enhance patient care, lower environmental risks, and encourage sustainable practices in healthcare organizations, by investigating these areas.

Staff expectations are also anticipated to change, just like those of patients and the general public. The growing demand for a workplace that supports employee wellbeing is advantageous to both the healthcare organization and the employee. It has been demonstrated that a safe and healthy work environment boosts productivity and employee retention; some studies even contend that a "greener" workplace results in fewer "sick days" being taken. Additionally, fostering employee well-being can help the company's reputation and draw in top talent. Prioritizing employee well-being is crucial for healthcare organizations if they want to have a long-lasting and prosperous future.

*4.2. Limitations of the Study*

There are some restrictions on the current study that can be addressed in subsequent research. These limitations are beyond the scope of this research. As a result of related institutional changes, this study is only conducted in Saudi Arabia. Future research can explore cultures within developing nations (e.g., Muslim countries, Arabic countries, and Islamic countries versus non-Islamic countries). This will help to clarify the variations and patterns in how businesses disclose their involvement in health initiatives and/or medical research in developing nations. The financial sector of the Saudi market was excluded from the study's sample. Future research can take Saudi financial sectors into account. The generalizability of the results to the entire Saudi market may be impacted by this restriction.

**5. Conclusions**

It is important to note that the following information is based on the most recent available information. The following are the results of a survey conducted by the National Institute of Standards and Technology (NIST) on the effectiveness of the standardized testing process. The study's findings reveal a statistically significant difference between non-financial companies' sustainable reporting and their HCSRD (i.e., participation in health-related programs). The Saudi government should take note of these companies' involvement in health-related programs, as a result of the variation in corporate sustainability reporting, in order to increase accountability and transparency for both the industry as a whole and sectors with low HCSRD.

Corporate sustainability reporting can build on the industry-specific metrics found in this study, to create a healthcare services sector supplement, that can address the sector's unique reporting requirements. Corporate sustainability reporting will subsequently encourage more healthcare service providers to investigate sustainability reporting within the framework for corporate sustainability reporting and improve the caliber of data provided in sustainability reports for organizations in this sector. The results of this study show

that sustainability reporting for hospitals is uncommon, and that few businesses in this industry think about using reporting indicators. Only a small percentage of reporting organizations and few businesses use corporate reporting guidelines when reporting their sustainability performance.

**Author Contributions:** Conceptualization, S.M.F.A. and K.M.A.; methodology, K.M.A.; software, K.M.A.; validation, A.M. and N.A.B.; formal analysis, K.M.A. and A.M.; investigation, S.M.F.A., K.M.A. and N.A.B.; resources, N.A.B.; data curation, S.M.F.A. and, K.M.A.; writing—original draft preparation, S.M.F.A.; writing—review and editing, S.M.F.A. and, K.M.A.; visualization, S.M.F.A.; supervision, S.M.F.A. and N.A.B.; project administration, N.A.B. and S.M.F.A.; funding acquisition, N.A.B. All authors have read and agreed to the published version of the manuscript.

**Funding:** This research was funded by Princess Nourah bint Abdulrahman University Researchers Supporting Project number (PNURSP2023R293), Princess Nourah bint Abdulrahman University, Riyadh, Saudi Arabia.

**Institutional Review Board Statement:** Not applicable.

**Informed Consent Statement:** Not applicable.

**Data Availability Statement:** Data available on the following link: https://www.my.gov.sa/wps/portal/snp/content/SDGPortal#header2_1, accessed on 31 December 2018.

**Acknowledgments:** The authors extend their appreciation to Princess Nourah bint Abdulrahman University Researchers Supporting Project, Princess Nourah bint Abdulrahman University, Riyadh, Saudi Arabia for funding this research work through the project number (PNURSP2023R293).

**Conflicts of Interest:** The authors declare no conflict of interest.

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
