# Peer review of "Exploring the Impact of Private Companies’ Participation in Health-Related Programs through Corporate Sustainable Reporting"

_sustainability, doi:10.3390/su15075906_

Round 1

Reviewer 1 Report (New Reviewer)

-       Minor English changes are required.

Some sentences are too long and can be divided into 2 or 3 sentences to make the paper easier to read.

-       From lines 92 to 95, there is a paragraph with no complete sentence. Please correct it.

Author Response

  •       Minor English changes are required.
  • Author response:
  • Thank you for the constructive review and feedback. English editing and review were done throughout the manuscript as per the reviewers' valuable recommendations.

Some sentences are too long and can be divided into 2 or 3 sentences to make the paper easier to read.

  • Author response:
  • Thank you for the constructive review and feedback. All required modifications were done as per the reviewers' valuable recommendations.
  •  
  •       From lines 92 to 95, there is a paragraph with no complete sentence. Please correct it.
  • Author response:
  • Thank you for the constructive review and feedback. All required modifications were done as per the reviewers' valuable recommendations.
  •  

Reviewer 2 Report (New Reviewer)

This paper is regular. Authors need to improve it with more references in the whole text, sections could be better with really deep analysis and connections, and some paragraphs could be clearer or without an impact. I recommend academic paper English-style revision. 

The authors must improve the paper's form and structure and clarify the contributions' focus. The concepts need to be revised and connected references.

My comments by sections are:

- Title: review the submission title and the pdf file. "Non-financial companies'"

-Abstract: Delete (1) Abstract, (2) Aim ..... It is a good finding, but the paper's real contribution and the literature review gap are unclear. 

-Introduction: Authors describe Saudi Arabian history, environment and motivation. This section needs more references and aligns with the theoretical framework. The final paragraph could be a short resume, section 2.... section 3.....

- Theoretical background: Paper has a CERES abstract and description. Try to balance with other reports or frameworks. This section needs the entire knowledge situation, possible gaps and contributions. This section is a description. 

Authors could align this section with building a sustainable healthcare framework. Try to analyse the four reasons, lines 216-221.

- Aim of the study: the questions are: how do previous sections support both hypotheses? Why do you select the sample and fieldwork?

-Methodology: the paper doesn't have this section. Consider this section and describe it step by step. And How and why select the phases. The authors present a mix between literature review, methodology and results in section 3. Health care analysis. 

This section does not clarify why authors use GLM, why not other methodologies, and how it supports the hypothesis. 

-Results: My comment is the same as the methodology. Authors need to support their analysis, hypothesis and contributions.

- Discussion: it is the only description. The authors could improve this section by considering aligning this section with the introduction and other sections. 

-Conclusions: It losses focus on the previous disorder. Align this section with the contribution and the research impact. It is not only a description. 

Author Response

This paper is regular. Authors need to improve it with more references in the whole text, sections could be better with really deep analysis and connections, and some paragraphs could be clearer or without an impact. I recommend academic paper English-style revision. 

Authors responses:

Thank you for the constructive review and feedback. The required English editing and review were done throughout the manuscript as per the reviewers' valuable recommendations.

The authors must improve the paper's form and structure and clarify the contributions' focus. The concepts need to be revised and connected references.

My comments by sections are:

- Title: review the submission title and the pdf file. "Non-financial companies'"

Authors responses:

Thank you for the constructive review and feedback. The title was reviewed and fixed as following: Exploring the impact of private companies’ participation in health-related programs through corporate sustainable reporting.

-Abstract: Delete (1) Abstract, (2) Aim ..... It is a good finding, but the paper's real contribution and the literature review gap are unclear. 

Authors responses:

Thank you for the constructive review and feedback. All the required modifications were done and the literature review gap was clarified in the abstract as following;

Background: Sustainability reporting has become a popular practice for corporations from various sectors all over the world, not only for its value as a means of tracking one's company's performance, but also as a tool for communicating performance to all involved stakeholders in any corporation. There is little research on how private companies participate in various health programs through maintaining efficient reporting; Aim: The study seeks to investigate the impact of private companies' participation in health-related programs via corporate sustainable reporting (or corporate social responsibility disclosures; CSRD). Methods: descriptive - correlational analysis, a balanced panel data of 117 Saudi listed businesses; Results: Corporate engagement in health care necessitates the identification of various hazards. As a result of the implementation of Saudi Arabia's 2030 Vision in 2016, the reporting of health-related CSRD (HCSRD) increased from 36% in 2015 to 48% in 2018. Conclusions: Corporate sustainable participation in healthcare can vary among firms indicating the different levels of influence in this regard. The Health Care sector having the lowest average amount of disclosures shows a lack of responsibility and control, as well as an inability to follow developments in the industry. The findings have practical implications for a range of stakeholders (e.g., regulators, investors, accounting professionals, and other institutions) of Health-related CSRD in Saudi Arabia.

-Introduction: Authors describe Saudi Arabian history, environment and motivation. This section needs more references and aligns with the theoretical framework. The final paragraph could be a short resume, section 2.... section 3.....

Authors responses:

Thank you for the constructive review and feedback. All the required modifications were done as per the valuable recommendations. Proper referencing was done and aligned with the theoretical framework. AAs well as, the required summarizing was done as following:

Criss et al.,(2019) found that financial conflict of interest (COI) sponsorship appears to be associated with a higher likelihood of studies reporting a benefit of robotic surgery and that the clinical results did not account for COI funding received (15). Another suggested that the relationships with industry offer opportunities for innovation, education, and research, but overlooking COI self-reporting may erode confidence in the academic integrity of the hand surgery literature (16). According to Ross et al. (2020), the use of financial incentives to reward primary health care providers (PCPs) for improving the quality of primary healthcare services is increasing, but there is little evidence to back this up. They also state that implementation should be done with caution, with a stronger theoretical foundation, a broader range of outcomes, and more extensive subgroup analysis (17).

- Theoretical background: Paper has a CERES abstract and description. Try to balance with other reports or frameworks. This section needs the entire knowledge situation, possible gaps and contributions. This section is a description. 

Authors responses:

Thank you for the constructive review and feedback. All the required modifications were done as the current study framework was linked with other reports and legitimacy theory clarifying the possible gaps and contributions as per the valuable guidance and recommendations.

Authors could align this section with building a sustainable healthcare framework. Try to analyse the four reasons, lines 216-221.

Authors responses:

Thank you for the constructive review and feedback. All the required modifications were done as the required alignment was done and the required analysis to the four reasons was added.

- Aim of the study: the questions are: how do previous sections support both hypotheses? Why do you select the sample and fieldwork?

Authors responses:

Thank you for the constructive review and feedback. All the required modifications were done as the required research question was clarified and the reasons for sample selection and filed work were clarified.

-Methodology: the paper doesn't have this section. Consider this section and describe it step by step. And How and why select the phases. The authors present a mix between literature review, methodology and results in section 3. Health care analysis. 

Authors responses:

Thank you for the constructive review and feedback. Actually, the methodology section was clarified and the results section to be clear in blue color. Both include all the recommended modifications.

This section does not clarify why authors use GLM, why not other methodologies, and how it supports the hypothesis. 

Authors responses:

Thank you for the constructive review and feedback. The reason of using GLM is clarified under the model specification and how it supports the hypothesis.

-Results: My comment is the same as the methodology. Authors need to support their analysis, hypothesis and contributions.

Authors responses:

Thank you for the constructive review and feedback. Actually, the results section was highlighted  in blue color to be clear as per the recommended modifications.

- Discussion: it is the only description. The authors could improve this section by considering aligning this section with the introduction and other sections. 

Authors responses:

Thank you for the constructive review and feedback. The discussion section was modified as per the reviewers valuable recommendations to be aligned with the other sections of the manuscript..

-Conclusions: It losses focus on the previous disorder. Align this section with the contribution and the research impact. It is not only a description. 

Authors responses:

Thank you for the constructive review and feedback. All required alignment was done with the previous order and the research impact. As well as the implications of the study clarify the implications of the study.

Reviewer 3 Report (New Reviewer)

Dear authors,

The research is interesting, it has a good structure, the topic is original. But, for me it was tiring to go through 5 introductory pages (which means 1/3 of research) and I did not found clear research objectives/research questions in this part of the paper. Then, the research methodology was clear, inclusive the sample construction. For me, Figure 1 was irrelevant. Additionally, when the two hypothesis were tested, it was not clear stated which of them is accepted or not.  

Author Response

The research is interesting, it has a good structure, the topic is original. But, for me it was tiring to go through 5 introductory pages (which means 1/3 of research) and I did not found clear research objectives/research questions in this part of the paper.

Authors responses:

Thank you for the constructive review and feedback. Actually, the length of the introduction section was required previously by the other reviewers to handle the study variables and the theoretical framework in details.

The research aim was highlighted in blue to be clarified and the required research question was added as per the valuable recommendations.

 Then, the research methodology was clear, inclusive the sample construction.

Authors responses:

Thank you for the constructive review and motivating feedback.

 For me, Figure 1 was irrelevant.

Authors responses:

Thank you for the constructive review and feedback. It was deleted.

Additionally, when the two hypothesis were tested, it was not clear stated which of them is accepted or not.  

Authors responses:

Thank you for the constructive review and feedback. The two-research hypothesis were accepted and it was cited inside the results section with table 6 and table 7.

Round 2

Reviewer 2 Report (New Reviewer)

The authors need to improve the paper structure, including their sections; focus on the following:

1. Literature review gap.

2. Research contribution. 

3. Research novelty. 

4. Knowledge generation. 

The topic has potential, but the paper doesn´t have these four requirements. 

Author Response

Responses to editor and reviewers’ comments

Dear Respected Professor Editor,

After warm greetings,

Kindly we would like to thank you and the respected reviewers on your valuable review and constructive feedback in order to improve the quality of our research paper.

Kindly find the responses on the respected reviewers’ comments and recommendations for improvement of the paper quality.

Reviewer 2 Comments:

The authors need to improve the paper structure, including their sections; focus on the following:

  1. Literature review gap.

We are thankful to the valuable review and constructive feedback and we need to clarify that the literature review was covered under the significance of reporting section as following and colored in blue in the manuscript;

Much research has focused on the aim of reporting since the voluntary adoption of sustainability reporting began in the 1980s and 1990s. According to Crane and Glozer (2016), the goals of sustainability and CSR communication are as follows: (1) stakeholder management to build relationships and influence behavior, (2) image enhancement to present the company in a favorable light, (3) legitimacy and accountability to signal appropriate and desirable activities, (4) consumer attitude and behavioral change, (5) sensemaking to communicate how the company and stakeholders make sense of their world, and (6) identity amplification. However, it has been stated that the overarching goal for which firms convey corporate sustainability and responsibility actions is to increase financial returns through participation in health-related programs. Indeed, studies and industry publications emphasize the financial benefit and value that sustainability reporting can provide to a company instead of investigating the impact of the corporates participation in different health-related programs and the sustainable reporting of these activities is in need for deep researches and investigation as there is limited reviews and gaps in fulfilling this research. [17]

  1. Research contribution. 

We are thankful to the valuable review and constructive feedback and we need to clarify that the research contribution was covered under the research contribution and implications section as following and colored in blue in the manuscript;

This study makes the following practical and scholarly contributions and implications. First, practically, many stakeholders can benefit from the current research. This is because sustainability reporting on non-profit organizations' engagement in health-related activities is fast changing, with several standards and frameworks emerging. Thus, this study may assist organizations' management in deciding which sustainability criteria and recommendations to follow while participating in health-related projects. As well as, Regulatory Agencies; currently there are no statutory requirements in Saudi Arabia for firms or the private sector to create and publish sustainability reports that include their participation in health-related activities. This research would aid in expanding policymakers’ understanding of placing relevant initiatives in place which perhaps encourage such sustainability reporting. Therefore, it also contributes to policymakers’ and interested researchers’ understanding of factors that could influence the HCSRD of developed and developing countries. In relation to Local Communities and Other Participants; this research provides insights for local communities where these companies operate, as well as other stakeholders such as employees, and social and environmental non-governmental organizations, about the adequacy and potentials of social sustainability reporting to meet their information needs and hold companies accountable. Also, companies yet to adopt sustainability reporting; this study will assist firms who have yet to implement sustainability reporting practices in understanding the benefits and drawbacks of this emerging reporting system and its influence on corporate performance. Firms will be in a better position to decide whether or not to implement this reporting scheme.

In relation to scholarly implications, academics might see the relevance of this research in the following ways: It would contribute to the enrichment of the literature on sustainability reporting of non-financial corporate participation in health-related programs, as well as provide students, scholars, and academics with more knowledge on the relationship between sustainability reporting and corporate performance in relation to health activities. Researchers would be able to refer to the study as a body of restricted knowledge.

Past literature has focused on the impact of global private equity funds' investments in health‐care systems and how financial instruments have induced the marketisation of the sector, leading to worsening health‐care provisions (13-14). The purpose of this article is to explore the impact of private sector/non-financial companies on participation in health-related programs through corporate sustainable reporting.

CERES pioneered the most widely used method of standardizing sustainability reporting (Coalition for Environmentally Responsible Economic). In 1998, the idea for a disclosure framework for sustainability information was born, and the Boston-based non-profit CERES launched the "Global Reporting Initiative" project with UNEP assistance (United Nations Environment Program).

  1. Research novelty. 

We are thankful to the valuable review and constructive feedback and we need to clarify that the research novelty was covered by the end of the significance of reporting section as following and colored in blue in the manuscript;

The research novelty returns to the study of the sustainable reporting in relation to health-related programs reporting and participation by the corporates which is not studied by the different previous researches as most of the previous researches focused on reporting the profit indicators of participation in different financial related activities.

  1. Knowledge generation. 

We are thankful to the valuable review and constructive feedback and we need to clarify that the knowledge generation was covered under the discussion and the conclusion sections as following and colored in blue in the manuscript;

Corporate sustainable participation in healthcare varies amongst non-financial firms indicating the different levels of influence in this regard. The Health Care sector performing among the lowest average level of disclosures (HCSRD) indicates a lack of accountability and control as well as slow in tracking respective changes in the industry. The results of this study show that there is a statistically significant variance between non-financial firms sustainable reporting with their HCSRD (i.e., participation in health related-programs). This variance in corporate sustainable reporting leads to the conclusion that the Saudi government should pay attention to the participation of these firms in health-related programs in order to improve respective accountability and transparency for sectors with low HCSRD and for the industry at large. Corporate sustainable reporting can build on the sector specific indicators discovered in this research to construct a healthcare services sector supplement that can fulfill the reporting needs and peculiarities of this sector. As a result, Corporate sustainable reporting will encourage additional healthcare service providers to explore sustainability reporting within the Corporate sustainable reporting framework, as well as increase the quality of information offered in sustainability reports for organizations in this sector. According to the findings of this study, sustainability reporting for hospitals is uncommon, and few firms in this sector consider reporting utilizing reporting indicators. Few firms use corporate reporting guidelines to report their sustainability performance, and only a small percentage of reporting organizations use corporate reporting rules. (29)

Future research opportunities include assessing the drivers and barriers to sustainability reporting, the implications of sustainability reporting on the healthcare service sector, how healthcare service providers react to the development of the GRI health services sector supplement for sustainability reporting, and the implications of sustainability reporting for public hospitals. Will CSR reporting improve patient care while reducing short and long-term risks to the environment and future human generations?

Furthermore, as with patients and the general public, staff expectations are expected to shift. Growing concerns for a well-being-promoting workplace benefit both the healthcare organization and the employee. A healthy and safe workplace environment has been shown to increase employee retention and productivity; studies even suggest that in a 'greener' workplace, fewer 'sick days' are taken (CABE, 2009).

The topic has potential, but the paper doesn´t have these four requirements. 

We are thankful to the valuable review and constructive feedback and recommendations. We hope that our responses will be accepted that time and suitable in answering the respected reviewers’ recommendations for our research quality improvement.

Thanks in advance for the valuable review and decision.

The corresponding author

Dr Sally M.Farghaly

Round 3

Reviewer 2 Report (New Reviewer)

Authors improved their previous version, but they need to reinforce more: Literature review gap, research contribution, research novelty, knowledge generation, and the English paper style.

It is still difficult to read.

Author Response

Reviewer 2 Comments:

Authors improved their previous version, but they need to reinforce more: Literature review gap, research contribution, research novelty, knowledge generation, and the English paper style.

It is still difficult to read.

Authors response:

Dear Respected Reviewer,

Kindly we thank you on your valuable feedback and constructive review to improve the quality of our research paper. We confirm that the recommended modifications were done and highlighted in yellow with full consideration to make all the paragraphs-related ideas clear with suitable and clear English. We hope that time, the paper will meet your valuable expectations.

Thanks in advance for all of your efforts

The corresponding author

This manuscript is a resubmission of an earlier submission. The following is a list of the peer review reports and author responses from that submission.

Round 1

Reviewer 1 Report

The article is devoted to studying the impact of the private sector / non-financial companies on participation in health-related programs through sustainable corporate reporting. The study's relevance is justified by the fact that financialization is defined as the increasing predominance of financial institutions and motives over the traditional non-financial sectors of the economy and ordinary people. Sociologists and political economists explore how financialization affects various social and economic life aspects, including labor relations and income inequality. The article conducted a descriptive-correlation study, balanced panel data of 117 Saudi non-financial companies on the stock exchange. The study results show that corporations' involvement in healthcare requires the identification of various risks. The disclosure of such an item improved from 36% in 2015 to 48% in 2018 due to the Saudi Arabia Vision 2030 implementation. In 2016 related programs with an indication of their impact on the corporation. The health sector with the lowest average number of pages in the report indicates a lack of accountability and oversight and a lack of ability to monitor changes in the sector.

Despite the satisfactory quality of the article, some shortcomings need to be corrected.

  1. The abstract should be rewritten. It is recommended to describe the actuality and aims of the paper briefly. The references should not be used in the abstract part.
  2. The data used for the experimental investigation should be described in more detail.
  3. The methods and models proposed by the authors should be highlighted and separated from known ones.
  4. It is recommended to compare obtained results with similar research in other countries.
  5. The scientific and practical novelty should be highlighted.
  6. The authors' contributions should be highlighted.
  7. The formatting should be fixed according to the template.

In summarizing my comments, I recommend that the manuscript is accepted after major revision. 

Author Response

Respected Professor Reviewer 

after warm greetings 

we are thankful to your constructive review and recommendations

English language and style

Yes

Can be improved

Must be improved

Not applicable

Is the content succinctly described and contextualized with respect to previous and present theoretical background and empirical research (if applicable) on the topic?

( )

(x)

( )

( )

Are all the cited references relevant to the research?

(x)

( )

( )

( )

Are the research design, questions, hypotheses and methods clearly stated?

( )

( )

(x)

( )

Are the arguments and discussion of findings coherent, balanced and compelling?

( )

( )

(x)

( )

For empirical research, are the results clearly presented?

( )

( )

(x)

( )

Is the article adequately referenced?

( )

(x)

( )

( )

Are the conclusions thoroughly supported by the results presented in the article or referenced in secondary literature?

( )

(x)

( )

( )

 ( ) Extensive editing of English language and style required
( ) Moderate English changes required
(x) English language and style are fine/minor spell check required

The required English editing was done all over the manuscript as recommended.

( ) I don't feel qualified to judge about the English language and style

Comments and Suggestions for Authors

The article is devoted to studying the impact of the private sector / non-financial companies on participation in health-related programs through sustainable corporate reporting. The study's relevance is justified by the fact that financialization is defined as the increasing predominance of financial institutions and motives over the traditional non-financial sectors of the economy and ordinary people. Sociologists and political economists explore how financialization affects various social and economic life aspects, including labor relations and income inequality. The article conducted a descriptive-correlation study, balanced panel data of 117 Saudi non-financial companies on the stock exchange. The study results show that corporations' involvement in healthcare requires the identification of various risks. The disclosure of such an item improved from 36% in 2015 to 48% in 2018 due to the Saudi Arabia Vision 2030 implementation. In 2016 related programs with an indication of their impact on the corporation. The health sector with the lowest average number of pages in the report indicates a lack of accountability and oversight and a lack of ability to monitor changes in the sector.

Despite the satisfactory quality of the article, some shortcomings need to be corrected.

  1. The abstract should be rewritten. It is recommended to describe the actuality and aims of the paper briefly. The references should not be used in the abstract part.

The abstract was rewrote as recommended including the aim and the references removed from the abstract.

  1. The data used for the experimental investigation should be described in more detail.

Detailed explanation to the data used was highlighted in blue as required

  1. The methods and models proposed by the authors should be highlighted and separated from known ones.

The model specification was clarified and highlighted in blue as recommended

  1. It is recommended to compare obtained results with similar research in other countries.

The recommended comparison was done as requested.

  1. The scientific and practical novelty should be highlighted.

The significance of the study and its scientific and practical novelty were clarified as recommended.

  1. The authors' contributions should be highlighted.

The authors contributions were clarified and highlighted as recommended

  1. The formatting should be fixed according to the template.

Formatting of the template was reviewed and fixed as recommended.

Reviewer 2 Report

This is a poorly written paper. The paper did not test any hypothesis, so I do not understand what the regression analysis was being used for.

The overall quality of the paper is bad and cannot be published in its current form

Author Response

Respected Professor Reviewer, 

After warm greetings, 

We are thankful to your constructive review 

English language and style

(x) Extensive editing of English language and style required

The required English editing was done all over the manuscript as recommended.

( ) Moderate English changes required
( ) English language and style are fine/minor spell check required
( ) I don't feel qualified to judge about the English language and style

Yes

Can be improved

Must be improved

Not applicable

Is the content succinctly described and contextualized with respect to previous and present theoretical background and empirical research (if applicable) on the topic?

( )

( )

(x)

( )

Are all the cited references relevant to the research?

( )

( )

(x)

( )

Are the research design, questions, hypotheses and methods clearly stated?

( )

( )

(x)

( )

Are the arguments and discussion of findings coherent, balanced and compelling?

( )

( )

(x)

( )

For empirical research, are the results clearly presented?

( )

( )

(x)

( )

Is the article adequately referenced?

( )

( )

(x)

( )

Are the conclusions thoroughly supported by the results presented in the article or referenced in secondary literature?

( )

( )

(x)

( )

Comments and Suggestions for Authors

This is a poorly written paper. The paper did not test any hypothesis, so I do not understand what the regression analysis was being used for.

The hypothesis was added as recommended.

The overall quality of the paper is bad and cannot be published in its current form

The paper was modified based on the reviewer’s recommendation and we hope that its quality will be improved and clarified after fixing all required changes.

Reviewer 3 Report

Good afternoon,

Please review the manuscript for typos and English language use. It is mandatory to use the same citation style in the manuscript (numbers in parentheses/ author names + year in parentheses). 

Some punctual observations below:
Line 38: `another factor on financialization is...` Please insert missing words.
Lines 43-45: Please begin the phrases with capital letters.
Line 48: What financial services do you refer to? Please explain.
Line 49: Capitalize first name of the author.
Define what PCP is before using the abbreviation.

Please use the theoretical section of the paper (you need to create one) to introduce the concepts used in the phrases below and explain how they relate, according to extant literature in the field:
`The purpose of this article is to explore the impact of private sector/non-financial companies on participation in health-related programs through corporate sustainable reporting. So that, the aim of the study is to explore the impact of private non-financial companies’ participation in health-related programs through corporate sustainable reporting.`

It is not clear why financialization is introduced as a concept in the paper. It does not seem to be useful for the research per se. Also, please review the key concepts. Taking into account the declared aim of the paper, the concepts seem to be more in line with `non-financial companies`, `health programs`, `medical research`,  `sustainable reporting`. The theoretical section shall clear those aspects and explain the intricacies.

The theoretical section shall discuss what private sector/ non-financial companies are, how they involve with health related programs, and how is that usually reflected in CSR reporting. The theoretical section also needs to explain what the impact of the participation in this type of programs is, how was the impact measured in previous studies, and how and why you decided to measure it the way you did. Please expand the bibliography and refer to more studies in this section. 

Please write about the SDG and Saudi Arabia's strategy for 2030 in the first part of the paper (Introduction), to explain the context. Please explain how it impacts private sector/ non-financial corporate participation in health care related programs.

The methodological section shall introduce the research objectives and/ or hypotheses, as well as the any details regarding the instruments you used to perform the analysis. 

The sample shall be presented after the research objectives and/ or hypotheses.

The discussion on the results needs to be expanded. Implications must be added. What do the results mean for non-financial companies in Saudi Arabia? What do they mean for the country's economy? How do the findings support the strategy for 2030? What are the implications of the findings for various sectors and for the society at large? Why are the results relevant for a reader from another part of the world/ what is the novelty factor?

Good luck!  

Author Response

Respected Professor Reviewer, 

After Warm greetings, 

We are thankful to your constructive review 

English language and style

( ) Extensive editing of English language and style required
(x) Moderate English changes required

The required English editing was done all over the manuscript as recommended.
( ) English language and style are fine/minor spell check required
( ) I don't feel qualified to judge about the English language and style

Yes

Can be improved

Must be improved

Not applicable

Is the content succinctly described and contextualized with respect to previous and present theoretical background and empirical research (if applicable) on the topic?

( )

( )

(x)

( )

Are all the cited references relevant to the research?

( )

(x)

( )

( )

Are the research design, questions, hypotheses and methods clearly stated?

( )

( )

(x)

( )

Are the arguments and discussion of findings coherent, balanced and compelling?

( )

( )

(x)

( )

For empirical research, are the results clearly presented?

( )

( )

(x)

( )

Is the article adequately referenced?

( )

( )

(x)

( )

Are the conclusions thoroughly supported by the results presented in the article or referenced in secondary literature?

( )

( )

(x)

( )

Comments and Suggestions for Authors

Good afternoon,

Please review the manuscript for typos and English language use. It is mandatory to use the same citation style in the manuscript (numbers in parentheses/ author names + year in parentheses). 

The references were reviewed and fixed as per the journal guidelines as recommended.

Some punctual observations below:
Line 38: `another factor on financialization is...` Please insert missing words.

The sentence was reviewed and fixed as recommended.

Lines 43-45: Please begin the phrases with capital letters.

The sentences were reviewed and fixed as recommended.

Line 48: What financial services do you refer to? Please explain.

The financial services were explained as recommended in details.

Line 49: Capitalize first name of the author.

The recommended changes were fixed.

Define what PCP is before using the abbreviation.

The full abbreviation was added as recommended

Please use the theoretical section of the paper (you need to create one) to introduce the concepts used in the phrases below and explain how they relate, according to extant literature in the field:
`The purpose of this article is to explore the impact of private sector/non-financial companies on participation in health-related programs through corporate sustainable reporting. So that, the aim of the study is to explore the impact of private non-financial companies’ participation in health-related programs through corporate sustainable reporting. `

The recommended changes were done and added in blue color.

It is not clear why financialization is introduced as a concept in the paper. It does not seem to be useful for the research per se. Also, please review the key concepts. Taking into account the declared aim of the paper, the concepts seem to be more in line with `non-financial companies`, `health programs`, `medical research`,  `sustainable reporting`. The theoretical section shall clear those aspects and explain the intricacies.

The required recommendations were added and fixed as recommended

The theoretical section shall discuss what private sector/ non-financial companies are, how they involve with health related programs, and how is that usually reflected in CSR reporting. The theoretical section also needs to explain what the impact of the participation in this type of programs is, how was the impact measured in previous studies, and how and why you decided to measure it the way you did. Please expand the bibliography and refer to more studies in this section. 
The required recommendations were added and fixed as recommended in relation to theoretical section

Please write about the SDG and Saudi Arabia's strategy for 2030 in the first part of the paper (Introduction), to explain the context. Please explain how it impacts private sector/ non-financial corporate participation in health care related programs.

The SDG and Saudi Arabian strategy for 2030 was added as recommended

The methodological section shall introduce the research objectives and/ or hypotheses, as well as the any details regarding the instruments you used to perform the analysis. 

The aim and the hypothesis were added in the methodological section as recommended.

The sample shall be presented after the research objectives and/ or hypotheses.

The required modifications were done as recommended

The discussion on the results needs to be expanded. Implications must be added. What do the results mean for non-financial companies in Saudi Arabia? What do they mean for the country's economy? How do the findings support the strategy for 2030? What are the implications of the findings for various sectors and for the society at large? Why are the results relevant for a reader from another part of the world/ what is the novelty factor?

The required modifications were done

Round 2

Reviewer 1 Report

Thanks for the authors for their analysis and considering some of the reviewer's comments and recommendations. Still, some drawbacks needs correction:

1. The methods section should be expanded, and the methodology should be described in more details.

2. Obtained results should be compared with more studies in a field.

3. The practical novelty is still not clear. It should be described in more details.

Author Response

Thanks for the authors for their analysis and considering some of the reviewer's comments and recommendations. Still, some drawbacks needs correction:

  1. The methods section should be expanded, and the methodology should be described in more details.

Author responses: we are thankful for the constructive and valuable feedback and the recommended modifications were done and highlighted in yellow for expanding the methodology section

  1. Obtained results should be compared with more studies in a field.
  2. Author responses: we are thankful for the constructive and valuable feedback and the recommended modifications were done and highlighted in yellow for comparing the results with global studies.

  1. The practical novelty is still not clear. It should be described in more details.

Author responses: we are thankful for the constructive and valuable feedback and the recommended modifications were done and highlighted in yellow in the implications section

Reviewer 3 Report

Dear authors,
I appreciate your additional work. However, the paper still misses its backbone. My suggestions were made so that you would be able to identify the backbone of the paper and present it in a compelling way.
The literature review section does not help much in understanding what the purpose of the research was. You have chosen to write about too many things in this paper: CSR reporting, financialization, sustainability, health-related projects, non-financial companies etc. This is way too much. It does not allow you to go in depth in any of those areas. The intricacies are still not clear.
The methodology is still not very robust. Perhaps more explanations are necessary, in a different paper, with a different topic!
The implications of the research are still unclear.
I am sorry to reject your paper. You need to prepare a solid theoretical background for the research, stemming from a solid theoretical ground and to explain what your findings really bring to the field/ society. Also, please bear in mind that papers are not broad works, such as books! Political economy and sociology have no real use in the context of a paper which investigates CSR reporting. Focus on one topic and explore it in depth. Renaissance-like theoretical undertakings do not fit ISI journals, unfortunately.
All the best,

Author Response

Dear authors,
I appreciate your additional work. However, the paper still misses its backbone. My suggestions were made so that you would be able to identify the backbone of the paper and present it in a compelling way.
The literature review section does not help much in understanding what the purpose of the research was. You have chosen to write about too many things in this paper: CSR reporting, financialization, sustainability, health-related projects, non-financial companies etc. This is way too much. It does not allow you to go in depth in any of those areas. The intricacies are still not clear.

Author responses: we are thankful for the constructive and valuable feedback and the recommended modifications were done and highlighted in yellow in the theoretical framework and introduction section.

The methodology is still not very robust. Perhaps more explanations are necessary, in a different paper, with a different topic!

Author responses: we are thankful for the constructive and valuable feedback and the recommended modifications were done and highlighted in yellow in methodology section

The implications of the research are still unclear.

Author responses: we are thankful for the constructive and valuable feedback and the recommended modifications were done and highlighted in yellow in the implications section

I am sorry to reject your paper. You need to prepare a solid theoretical background for the research, stemming from a solid theoretical ground and to explain what your findings really bring to the field/ society. Also, please bear in mind that papers are not broad works, such as books! Political economy and sociology have no real use in the context of a paper which investigates CSR reporting. Focus on one topic and explore it in depth. Renaissance-like theoretical undertakings do not fit ISI journals, unfortunately.

Author responses: we are thankful for the constructive and valuable feedback and we appreciate all of your efforts to help us to improve the quality of our research. We hope that time, it will be improved based on your valuable recommendations that we are foolowing.

All the best,

Round 3

Reviewer 3 Report

Dear authors,

I appreciate you adding information to the paper. I now see the focus shifts towards sustainability reporting - which is ok - yet the abstract and the theoretical section mix other influences and point to other research directions, and the research methodology is still not robust. Thus, the paper fails to bring to the fore significant useful information to the readers (i.e. its relevance for the academic community is low and the implications of the research results are not too obvious). Once again, I recommend transforming the paper into a new one, in which financialization, psychology, and sociology have no place, and focus, instead, on the stream of sustainability reporting and the impact it has on the company. The research needs to be greatly improved, starting with the purpose, hypotheses, and methodology.
Wishing you all the best and inspiration!

Author Response

Dear Respected Professor,

After warm greetings,

Kindly we would like to thank you on your valuable feedback and continuous support to improve the quality of our research paper. The authors thank you for your constrictive evaluation and review.

The abstract and the theoretical section were modified and highlighted based on your valuable recommendations.

The research methodology including the aim, hypotheses and the field work were modified as recommended and highlighted within the manuscript.

All recommended changes of removing unrequired concepts were done.

Implications of the study and future researches were reviewed and modified as recommended for the benefit and interest of the readers.

We hope that time our manuscript will get acceptance based on following the recommended modifications and trying to fix it by doing our best.

Sincerely,

Corresponding Author,

Dr Sally Mohammed Farghaly
